# A Game-Theoretic Approach to Recommendation Systems with Strategic Content Providers

**Omer Ben-Porat** and **Moshe Tennenholtz**
Technion - Israel Institute of Technology
Haifa 32000 Israel
{omerbp@campus,moshe@ie}.technion.ac.il

## Abstract

We introduce a game-theoretic approach to the study of recommendation systems with strategic content providers. Such systems should be fair and stable. Showing that traditional approaches fail to satisfy these requirements, we propose the Shapley mediator. We show that the Shapley mediator fulfills the fairness and stability requirements, runs in linear time, and is the only economically efficient mechanism satisfying these properties.

## 1 Introduction

Recommendation systems (RSs hereinafter) have rapidly developed over the past decade. By predicting a user preference for an item, RSs have been successfully applied in a variety of applications. Moreover, the amazing RSs offered by giant e-tailers and e-marketing platforms, such as Amazon and Google, lie at the heart of online commerce and marketing on the web. However, current significant challenges faced by personal assistants (e.g. Cortana, Google Now and Alexa) and mobile applications go way beyond the practice of predicting the satisfaction levels of a user from a set of offered items. Such systems have to generate recommendations that satisfy the needs of both the end users and other parties or stakeholders [8, 39]. Consider the following cases:

• When Alice drives her car, her personal assistant runs the default navigation application. When she makes a stop at a junction, the personal assistant may show Alice advertisements provided by neighborhood stores, or an update on the stock market status as provided by financial brokers. Each of these pieces of information — the plain navigation content, the local advertisements and the financial information — are served by different content providers. These content providers are all competing over Alice's attention at a given point. The personal assistant is aware of Alice's satisfaction with each content, and needs to select the right content to show at a particular time.

• Bob is reading news of the day on his mobile application. The application, aware of Bob's interests, is presenting news deemed most relevant to him. The news is augmented by advertisements, provided by competing content providers, as well as articles by independent reporters. The mobile application, balancing Bob's taste and the interests of the content providers, determines the mix of content shown to Bob.

In these contexts, the RS integrates information from various providers, often sponsored content, which is probably relevant to the user. The content providers are *strategic* — namely, make decisions based on the way the RS operates, aiming at maximizing their exposure. For instance, to draw Bob's attention, a content provider strategically selects the topic of her news item, aiming at maximizing the exposure to her item. On the one hand, fair content provider treatment is critical for smooth efficient use of the system and also to maintained content provider engagement over time. On the other hand, the strategic behavior of the content providers may lead to instability of the system: a content provider

might choose to adjust the content she offers in order to increase the expected number of displays to the users, assuming the others stick to their offered contents.

In this paper, we study ways of overcoming this dilemma using canonical concepts in game theory to impose two requirements on the RS: fairness and stability. Fairness is formalized as the requirement of satisfying fairness-related properties, and stability is defined as the existence of a pure Nash equilibrium. Analyzing RSs that satisfy these two requirements is the main goal of this paper.

Our first result is that traditional RSs fail to satisfy both of the above requirements. Traditional RSs are complete, in the sense that they always show some content to the user, but it turns out that this completeness property cannot be satisfied simultaneously with the fairness and equilibrium existence requirements. This impossibility result is striking and calls for a search of a fair and stable RS. To do so, we model the setting as a cooperative game, binding content provider payoffs with user satisfaction. We resort to a core solution concept in cooperative game theory, the Shapley value [35], which is a celebrated mechanism for value distribution in game-theoretic contexts (see, e.g., [28]). In our work, it is proposed as a tool for recommendations, namely for setting display probabilities. Since the Shapley value is employed in countless settings for fair allocation, it is not surprising that it satisfies our fairness properties. In addition, we prove that the related RS, termed the *Shapley mediator*, does satisfy the stability requirement. In particular, we show that the Shapley mediator possesses a potential function [27], and therefore any better-response learning dynamics converge to an equilibrium (see, e.g., [13, 17]). Note that this far exceeds our minimal stability requirement from the RS.

Implementation in commercial products would require the mediator to be computationally tractable. The mediator interacts with users; hence a fast response is of great importance. In another major result, we show that the Shapley mediator has a computationally efficient implementation. The latter is in contrast to the intractability of the Shapley value in classical game-theoretic contexts [14]. Another essential property of the Shapley mediator is economic efficiency [36]. Unlike cooperative games, where the Shapley value can be characterized as the only solution concept to satisfy properties equivalent to fairness and economic efficiency, in our setting the Shapley mediator is not characterized solely by fairness and economic efficiency. Namely, one can find other simple mediators that satisfy these two properties. However, we show that the Shapley mediator is the unique mediator to satisfy the fairness, economic efficiency and stability requirements. Importantly, our study stems from a rigorous definition of the minimal requirements from an RS, and so characterizes a unique RS. Interested in understanding the ramification on user utility, we introduce a rigorous analysis of user utility in (strategic) recommendation systems, and show that the Shapley mediator is not inferior to traditional approaches.

Due to space constraints, the proofs are deferred to the full version [5].

## 1.1   Related work

This work contributes to three interacting topics: fairness in general machine learning, multi-stakeholder RSs and game theory.

The topic of fairness is receiving increasing attention in machine learning [6, 12, 30, 32] and data mining [23]. A major line of research is discrimination aware classification [16, 19, 21, 38], where classification algorithms must maintain high predictive accuracy without discriminating on the basis of a variable representing membership in a protected class, e.g. ethnicity. In the context of RSs, the work of Kamishima et al. [20, 22] addresses a different aspect of fairness (or lack thereof): bias towards popular items. The authors propose a collaborative filtering model which takes into account viewpoints given by users, thereby tackling the tendency for popular items to be recommended more frequently, a problem posed in [29]. A related problem is over-specialization, i.e., the tendency to recommend items similar to those already purchased or liked in the past, which is addressed in [1].

Zheng [39] surveys multi-stakeholder RSs, and highlights practical applications. Examples include RSs for sharing economies (e.g. AirBnB, Uber, etc.), online dating [31], and recruiting [37]. Burke [8] discusses fairness in multi-stakeholder RSs, and presents a taxonomy of classes of fairness-aware RSs. The author distinguishes between user fairness, content provider fairness and pairwise fairness, and reviews applications for these fairness types. A practical problem concerning fairness in multi-stakeholder RSs is discussed in [26]. In their work, an online platform is used by users who play two roles: customers seeking recommendations and content providers aiming for exposure. They report,

Table 1: Consider an arbitrary game, a fixed strategy profile $\boldsymbol{X}$ and an arbitrary user $u_i$. TOP selects uniformly among the players that satisfy $u_i$ the most. The Bradley-Terry-Luce mediator [7, 25], or simply BTL, selects player $j$ w.p. proportional to her satisfaction level over the sum of satisfaction levels. NONE displays no item, and RAND selects uniformly among players with a positive satisfaction level. Both TOP and BTL satisfy $\mathbf{F}$, but do not satisfy $\mathbf{S}$. NONE and RAND satisfy $\mathbf{S}$, but do not satisfy $\mathbf{F}$. The bottom line refers to the Shapley mediator, SM, which is defined and analyzed in Section 3. In contrast to the other mediators, SM satisfies both $\mathbf{F}$ and $\mathbf{S}$.

| MEDIATOR | PROBABILITY COMPUTATION $\mathbb{P}\left(\mathcal{M}(\boldsymbol{X}, u_i) = j\right)$ | FAIRNESS ($\mathbf{F}$) | STABILITY ($\mathbf{S}$) |
|---|---|---|---|
| TOP | $\dfrac{\mathbb{1}_{j \in \arg\max_{j'} \sigma_i(X_{j'})}}{\left|\arg\max_{j'} \sigma_i(X_{j'})\right|}$ | $\checkmark$ | $\times$ (THEOREM 1) |
| BTL | $\dfrac{\sigma_i(X_j)}{\sum_{j'=1}^{N} \sigma_i(X_{j'})}$ | $\checkmark$ | $\times$ (THEOREM 1) |
| NONE | $0$ | $\times$ | $\checkmark$ |
| RAND | $\dfrac{\mathbb{1}_{\sigma_i(X_j)>0}}{\sum_{j'=1}^{N} \mathbb{1}_{\sigma_i(X_{j'})>0}}$ | $\times$ | $\checkmark$ |
| SM (SECTION 3) | EQUATION (2) | $\checkmark$ | $\checkmark$ (THEOREM 2) |

based on empirical evidence, that collaborative filtering techniques tend to create rich-gets-richer scenarios, and propose a method for re-ranking scores, in order to improve exposure distribution across the content providers.

Note that all the work above considers traditional machine learning tasks that enforce upon the solution some form of fairness, as defined specifically for each task. They suggest additional considerations, but do not consider that the parties (i.e., users, content providers) will change their behavior as a result of the new mechanism, nor examine the game theoretic aspects imposed by the selection of the RS in a formal manner. To the best of our knowledge, our work is the first to suggest a fully grounded approach to content provider fairness in RSs.

Finally, strategic aspects of classical machine learning tasks were also introduced recently [3, 4]. The idea that a recommendation algorithm affects content-provider policy, and as a result must be accompanied by a game-theoretic study is key to recent works in search/information retrieval [2, 33]; so far, however, such work has not dealt with the issue of fairness.

## 2 Problem formulation

From here on, our ideas will be exemplified in the following motivational example: a mobile application (or simply app) is providing users with valuable content. A set of players (advertisers) publish their items (advertisements) on the app. When a user enters the app, a mediator (RS/advertising engine) decides whether to display an item to that user or not, and which player's item to display. The reader should notice that while we use that motivation for the purpose of exposition, our formal model and results are applicable to a whole range of RSs with strategic content providers.

Formally, the recommendation game is defined as follows:

- A set of users $\mathcal{U} = \{u_1, \ldots, u_n\}$, a set of players $[N] = \{1, \ldots N\}$, and a mediator $\mathcal{M}$.

- The set of items (e.g. possible ad formats/messages to select from) available to player $j$ is denoted by $\mathcal{L}_j$, which we assume to be finite. A *strategy* of player $j$ is an item from $\mathcal{L}_j$.

- Each user $u_i$ has a satisfaction function $\sigma_i : \mathcal{L} \to [0, 1]$, where $\mathcal{L} = \bigcup_{j=1}^{N} \mathcal{L}_j$ is the set of all available items. In general, $\sigma_i(l)$ measures the *satisfaction level* of $u_i$ w.r.t. $l$.

- When triggered by the app, $\mathcal{M}$ decides which item to display, if any. Formally, given the strategy profile $\boldsymbol{X} = (X_1, \ldots, X_N)$ and a user $u_i$, $\mathcal{M}(\boldsymbol{X}, u_i)$ is a distribution over $[N] \cup \{\emptyset\}$, where $\emptyset$ symbolizes maintaining the plain content of the app. That is, displaying no item at all. We refer to

$\mathbb{P}\left(\mathcal{M}(\boldsymbol{X}, u_i) = j\right)$ as the probability that player $j$'s item will be displayed to $u_i$ under the strategy profile $\boldsymbol{X}$.

- Each player gets one monetary unit when her item is displayed to a user. Therefore, the expected payoff of player $j$ under the strategy profile $\boldsymbol{X}$ is $\pi_j(\boldsymbol{X}) = \sum_{i=1}^{n} \mathbb{P}\left(\mathcal{M}(\boldsymbol{X}, u_i) = j\right)$.
- The *social welfare* of the players under the strategy profile $\boldsymbol{X}$ is the expected number of displays, $V(\boldsymbol{X}) = \sum_{j=1}^{N} \pi_j(\boldsymbol{X})$.

For ease of notation, we shall sometimes refer to $\sigma_i(\boldsymbol{X})$ as the maximum satisfaction level of user $u_i$ from the items in $\boldsymbol{X}$, i.e., $\sigma_i(\boldsymbol{X}) = \max_j \sigma_i(X_j)$.

We demonstrate our setting with the following example.

**Example 1.** Consider a game with two players and three users. Let $\mathcal{L}_1 = \{l_1, l_2\}, \mathcal{L}_2 = \{l_3\}$ such that the satisfaction levels of the users with respect to the items are

$$
\begin{array}{c c c c}
 & u_1 & u_2 & u_3 \\
\begin{matrix} l_1 \\ l_2 \\ l_3 \end{matrix} & \left[ \begin{matrix} 0.1 & 0.9 & 0.2 \\ 0.8 & 0.7 & 0.9 \\ 0.9 & 0.8 & 0.1 \end{matrix} \right.
\end{array}.
$$

Consider a mediator displaying each user with the most satisfying item to her taste, denoted by TOP. For example, $\mathbb{P}(\text{TOP}\,((l_1, l_3), u_1) = 2) = 1$, since $\sigma_1(l_3) = 0.9 > \sigma_1(l_1) = 0.1$. The profile $(l_1, l_3)$ will probably be materialized in realistic scenarios, since the payoff of player 1 under the strategy profile $(l_2, l_3)$ is $\pi_1(l_2, l_3) = 1$, while $\pi_1(l_1, l_3) = 2$. Notice that from the users' perspective,[1] this profile is not optimal, since $\sum_{i=1}^{3} \sigma_i(l_1, l_3) = 0.9 + 0.9 + 0.2 = 2$, while $\sum_{i=1}^{3} \sigma_i(l_2, l_3) = 2.6$; hence, the users suffer from strategic behavior of the players.

After defining general recommendation games, we now present a few properties that one may desire from a mediator. First and foremost, a mediator has to be *fair*. The following is a minimal set of fairness properties:

**Null Player**. If $\sigma_i(X_j) = 0$, then it holds that $\mathbb{P}\left(\mathcal{M}(\boldsymbol{X}, u_i) = j\right) = 0$. Informally, an item will not be displayed to $u_i$ if it has zero satisfaction level w.r.t. him.

**Symmetry**. If $u_i$ has the same satisfaction level from two items, they will be displayed with the same probability. Put differently, if $\sigma_i(X_j) = \sigma_i(X_m)$, then $\mathbb{P}\left(\mathcal{M}(\boldsymbol{X}, u_i) = j\right) = \mathbb{P}\left(\mathcal{M}(\boldsymbol{X}, u_i) = m\right)$.

**User-Independence**. Given the selected items, the display probabilities depend only on the user: if user $u_{i'}$ is removed from/added to $\mathcal{U}$, $\mathbb{P}\left(\mathcal{M}(\boldsymbol{X}, u_i) = j\right)$ will not change, i.e.,

$$\mathbb{P}\left(\mathcal{M}(\boldsymbol{X}, u_i) = j\right) = \mathbb{P}\left(\mathcal{M}(\boldsymbol{X}, u_i) = j \mid u_{i'} \in \mathcal{U}\right).$$

**Leader Monotonicity**. $\mathcal{M}$ displays the most satisfying items (w.r.t. a specific user) with higher probability than it displays other items. Formally, if $j \in \arg\max_{j' \in [N]} \sigma_i(X_{j'})$ and $m \notin \arg\max_{j' \in [N]} \sigma_i(X_{j'})$, then $\mathbb{P}\left(\mathcal{M}(\boldsymbol{X}, u_i) = j\right) > \mathbb{P}\left(\mathcal{M}(\boldsymbol{X}, u_i) = m\right)$.

For brevity, we denote the above set of fairness properties by **F**. In addition, an essential property in a system with self-motivated participants is that it will be stable. Instability in such systems is a result of a player aiming to improve her payoff given the items selected by others. A minimal requirement in this regard is stability against unilateral deviations as captured by the celebrated pure Nash equilibrium concept, herein denoted PNE. A strategy profile $\boldsymbol{X} = (X_1, \dots, X_N)$ is called a *pure Nash equilibrium* if for every player $j \in [N]$ and any strategy $X_j' \in \mathcal{L}_j$ it holds that $\pi_j(X_j, \boldsymbol{X}_{-j}) \geq \pi_j(X_j', \boldsymbol{X}_{-j})$, where $\boldsymbol{X}_{-j}$ denotes the vector $\boldsymbol{X}$ of all strategies, but with the $j$-th component deleted. We use the notion of PNE to formalize the stability requirement:

**Stability**. Under any set of players, available items, users and user satisfaction functions, the game induced by $\mathcal{M}$ possesses a PNE.

For brevity, we denote this property by **S**. The goal of this paper is to devise a computationally tractable mediator that satisfies both **F** and **S**.[2]

## 2.1 Impossibility of classical approaches

We highlight a few benchmark mediators in Table 1, including TOP, which was introduced informally in Example 1. Another interesting mediator is BTL, which follows the lines of the Bradley-Terry-Luce model [7, 25]. BTL is addressed here as a representative of a wide family of weight-based mediators: mediators that distribute display probability according to weights, determined by a monotonically increasing function of the user satisfaction (e.g., softmax). Common to TOP, BTL and any other weight-based mediator, is that an item is displayed to a user with probability 1.[3] We model this property as follows.

**Complete**. For any recommendation game and any strategy profile $X$, $\sum_{j=1}^{N} \mathbb{P}\left(\mathcal{M}(X, u_i) = j\right) = 1$.

Since the goal of an RS is to provide useful content to users, satisfying **Complete** seems justified. Although it seems unreasonable to avoid showing any content to a certain user at a certain time, it turns out that this avoidance is crucial in order to satisfy our requirements.

**Theorem 1.** *No mediator can satisfy* $F$, $S$ *and* **Complete**.

*Proof sketch.* We construct a game with two players, three users and three strategies, and show that no mediator can satisfy $F$, $S$ and **Complete**. Importantly, our technique can be used to show that any arbitrary game does not possess a PNE or that a slight modification of this game does not possess a PNE.

Consider the following satisfaction matrix:

$$
\begin{array}{c}
\\
l_1 \\
l_2 \\
l_3
\end{array}
\begin{array}{ccc}
u_1 & u_2 & u_3 \\
\left[\begin{array}{ccc}
0 & y & x \\
x & 0 & y \\
y & x & 0
\end{array}\right],
\end{array}
$$

where $(x, y) \in (0, 1]^2$. Let $\mathcal{L}_1 = \mathcal{L}_2 = \{l_1, l_2, l_3\}$ (i.e., a symmetric two-player game). By using the properties of $F$ we characterize the structure of the induced normal form game. We show that in this normal form game, a PNE only exists if $\mathbb{P}\left(\mathcal{M}((l_2, l_3), u_1) = 1\right) = 0.5$ (and similarly to the other users and strategy profiles, due to **User-Independence**). Since this holds for every $x$ and $y$, the mediator displays a random item for each user under any strategy profile. Recall that a random selection does not satisfy **Leader Monotonicity**; hence, no mediator can satisfy $F$, $S$ and **Complete**. $\qquad\square$

Moreover, Theorem 1 is not sensitive to the sum of the display probabilities being equal to 1. One can show a similar argument for any mediator that displays items with constant probabilities, i.e., $\sum_{j=1}^{N} \mathbb{P}\left(\mathcal{M}(X, u_i) = j\right) = c$ for some $0 < c \leq 1$. Theorem 1 suggests that $\sum_{j=1}^{N} \mathbb{P}\left(\mathcal{M}(X, u_i) = j\right)$ should be bounded to the user satisfaction levels. In the next section, we show a novel way of doing so.

## 3 Our approach: the Shapley mediator

In order to provide a fair and stable mediator, we resort to cooperative game theory. Informally, a cooperative game consists of two elements: a set of players $[N]$ and a characteristic function $v : 2^{[N]} \rightarrow \mathbb{R}$, where $v$ determines the value given to every coalition, i.e., every subset of players. The analysis of cooperative games focuses on how the collective payoff of a coalition should be distributed among its members.

One core solution concept in cooperative game theory is the Shapley value [35].

**Definition 1** (Shapley value). *Let* $(v, [N])$ *be a cooperative game such that* $v(\emptyset) = 0$. *According to the Shapley value, the amount that player* $j$ *gets is*

$$
\frac{1}{N!} \sum_{R \in \Pi([N])} \left(v(P_j^R \cup \{j\}) - v(P_j^R)\right), \tag{1}
$$

*where $\Pi([N])$ is the set of all permutations of $[N]$ and $P_j^R$ is the set of players in $[N]$ which precede player $j$ in the permutation $R$.*

One way to describe the Shapley value, is by imagining the process in which coalitions are formed: when player $j$ joins coalition $\mathcal{C}$, she demands her contribution to the collective payoff of the coalition, namely $v(\mathcal{C} \cup \{j\}) - v(\mathcal{C})$. Equation (1) is simply summing over all such possible demands, assuming that all coalitions are equally likely to occur.

For our purposes, we fix a strategy profile $\boldsymbol{X}$, and focus on an arbitrary user $u_i$. How should a mediator assign the probabilities of being displayed in a fair fashion? The *induced cooperative game* contains the same set of players. For every $\mathcal{C} \subseteq [N]$, let $\boldsymbol{X}_\mathcal{C}$ denote the strategy profile where all players missing from $\mathcal{C}$ are removed. We define the characteristic function of the induced cooperative game as

$$v_i(\mathcal{C}; \boldsymbol{X}) = \sigma_i(\boldsymbol{X}_\mathcal{C}),$$

where $\sigma_i(\boldsymbol{X}_\mathcal{C})$ is the maximal satisfaction level a user $u_i$ may obtain from the items chosen by the members of $\mathcal{C}$. Indeed, this formulation represents a collaborative behavior of the players, when they aim to maximize the satisfaction of $u_i$. Observe that $v_i(\cdot; \boldsymbol{X}) : 2^{[N]} \rightarrow \mathbb{R}$ is a valid characteristic function, hence $(v_i(\cdot; \boldsymbol{X}), [N])$ is a well defined cooperative game. Note that the selection of a mediator fully determines the probability of the events $\mathcal{M}(\boldsymbol{X}, u_i) = j$, and vice versa. The mediator that sets the probability of the event $\mathcal{M}(\boldsymbol{X}, u_i) = j$ according to the Shapley value of the induced cooperative game $(v_i(\cdot; \boldsymbol{X}), [N])$ is hereinafter referred to as *the Shapley mediator*, or SM for abbreviation.

### 3.1 Properties of the Shapley mediator

Since the Shapley value is employed in countless settings for fair allocation, it is not surprising that it satisfies our fairness properties.

**Proposition 1.** SM *satisfies* **F**.

We now show that recommendation games with SM possess a PNE. This is done using the notion of potential games [27]. A non-cooperative game is called *an exact potential game* if there exists a function $\Phi : \prod_j \mathcal{L}_j \rightarrow \mathbb{R}$ such that for any strategy profile $\boldsymbol{X} = (X_1, \ldots, X_N) \in \prod_j \mathcal{L}_j$, any player $j$ and any strategy $X'_j \in \mathcal{L}_j$, whenever player $j$ switches from $X_j$ to $X'_j$, the change in her payoff function equals the change in $\Phi$, i.e.,

$$\Phi(X_j, \boldsymbol{X}_{-j}) - \Phi(X'_j, \boldsymbol{X}_{-j}) = \pi_j(X_j, \boldsymbol{X}_{-j}) - \pi_j(X'_j, \boldsymbol{X}_{-j}).$$

This brings us to the main result of this section:

**Theorem 2.** *Recommendation games with the Shapley mediator are exact potential games.*

Thus, due to Monderer and Shapley [27], any recommendation game with the Shapley mediator possesses at least one PNE, and the set of pure Nash equilibria corresponds to the set of argmax points of the potential function; therefore, SM satisfies **S**.

**Corollary 1.** SM *satisfies* **S**.

In fact, Theorem 2 proves a much stronger claim than merely the existence of PNE. A better-response dynamics is a sequential process, where in each iteration an arbitrary player unilaterally deviates to a strategy which increases her payoff.

**Corollary 2.** *In recommendation games with the Shapley mediator, any better-response dynamics converges.*

This convergence guarantee allows the players to learn which items to pick in order to maximize their payoffs. Indeed, as has been observed by work on the topic of online recommendation and advertising systems (e.g. sponsored search [10]), convergence to PNE is essential for system stability, as otherwise inefficient fluctuations may occur.

## 4 Linear time implementation

In Section 3 we showed that the Shapley mediator, SM, satisfies **F** and **S**. Therefore, it fulfills our requirements stated in Section 2. However, implementation in commercial products would require

the mediator to be computationally tractable. The mediator interacts with users; hence a fast response is of great importance. In general, since Equation (1) includes $2^N$ summands, the computation of the Shapley value in a cooperative game need not be tractable. Indeed, the computation often involves marginal contribution nets [11, 18]. In the following theorem we derive a closed-form formula for calculating the display probabilities under the Shapley mediator, which allows it to compute the display probabilities in linear time.

**Theorem 3.** *Let $\boldsymbol{X}$ be a strategy profile, and let $\sigma_i^m(\boldsymbol{X})$ denote the $m$'th entry in the result of sorting $(\sigma_i(X_1), \ldots, \sigma_i(X_N))$ in ascending order, preserving duplicate elements. The Shapley mediator displays player $j$'s item to a user $u_i$ with probability*

$$\mathbb{P}\left(\mathrm{SM}(\boldsymbol{X}, u_i) = j\right) = \sum_{m=1}^{\rho_i^j(\boldsymbol{X})} \frac{\sigma_i^m(\boldsymbol{X}) - \sigma_i^{m-1}(\boldsymbol{X})}{N - m + 1}, \tag{2}$$

*where $\sigma_i^0(\boldsymbol{X}) = 0$, and $\rho_i^j(\boldsymbol{X})$ is an index such that $\sigma_i(X_j) = \sigma_i^{\rho_i^j(\boldsymbol{X})}(\boldsymbol{X})$.*

The Shapley mediator is implemented in Algorithm 1. As an input, it receives a strategy profile and a user, or equivalently user satisfaction levels from that strategy profile. It outputs a player's item with a probability equal to her Shapley value in the cooperative game defined above. Note that the run-time of Algorithm 1 is linear in the number of players, i.e., $\mathcal{O}(N)$. A direct result from Theorem 3 and **User-Independence** (see Section 2) is that player payoffs can be calculated efficiently.

**Corollary 3.** *In recommendation games with the Shapley mediator, the payoff of player $j$ under the strategy profile $\boldsymbol{X}$ is given by $\pi_j(\boldsymbol{X}) = \sum_{i=1}^n \sum_{m=1}^{\rho_i^j(\boldsymbol{X})} \frac{\sigma_i^m(\boldsymbol{X}) - \sigma_i^{m-1}(\boldsymbol{X})}{N - m + 1}$.*

To facilitate understanding of the Shapley mediator and its fast computation, we reconsider Example 1 above.

**Example 2.** Consider the game given in Example 1. According to the Shapley mediator, the display probabilities of player 1 under the strategy profile $\boldsymbol{X} = (l_2, l_3)$ are

$$\mathbb{P}(\mathrm{SM}\left(\boldsymbol{X}, u_1\right) = 1) = \frac{\sigma_1^1\left(\boldsymbol{X}\right) - \sigma_1^0\left(\boldsymbol{X}\right)}{2} = \frac{0.8 - 0}{2} = 0.4,$$

$$\mathbb{P}(\mathrm{SM}\left(\boldsymbol{X}, u_2\right) = 1) = \frac{\sigma_2^1\left(\boldsymbol{X}\right) - \sigma_2^0\left(\boldsymbol{X}\right)}{2} = \frac{0.7 - 0}{2} = 0.35,$$

$$\mathbb{P}(\mathrm{SM}\left(\boldsymbol{X}, u_3\right) = 1) = \frac{\sigma_3^1\left(\boldsymbol{X}\right) - \sigma_3^0\left(\boldsymbol{X}\right)}{2} + \frac{\sigma_3^2\left(\boldsymbol{X}\right) - \sigma_3^1\left(\boldsymbol{X}\right)}{1} = \frac{0.1 - 0}{2} + \frac{0.9 - 0.1}{1} = 0.85.$$

It follows that $\pi_1(l_2, l_3) = \frac{8}{5}$ while $\pi_1(l_1, l_3) = \frac{7}{10}$, and the profile to be materialized is $(l_2, l_3)$. Indeed, it can be verified that this is the unique PNE of the corresponding game. Moreover, while the unique PNE under TOP (see Example 1 in Section 2) results in a user utility of 2, the unique PNE under the Shapley mediator results in user utility of

$$\sum_{i=1}^3 (\sigma_i(l_2)\mathbb{P}(\mathrm{SM}\left((l_2, l_3), u_i\right) = 1)) + (\sigma_i(l_3)\mathbb{P}(\mathrm{SM}\left((l_2, l_3), u_i\right) = 2)) = 2.145 > 2.$$

Hence, the users benefit from the Shapley mediator is greater than from the TOP mediator. This is in addition to the main property of the Shapley mediator, probabilistic selection according to the central measure of fair allocation.

## 5 Uniqueness of the Shapley mediator

As analyzed in Subsection 2.1, Theorem 1 suggests that a mediator cannot satisfy both **F** and **S** if it sets the probabilities such that $\sum_{j=1}^N \mathbb{P}\left(\mathcal{M}(\boldsymbol{X}, u_i) = j\right)$ is constant. One way of determining $\sum_{j=1}^N \mathbb{P}\left(\mathcal{M}(\boldsymbol{X}, u_i) = j\right)$ is defined as follows.

**Efficiency**. The probability of displaying an item to $u_i$ is the maximal satisfaction level $u_i$ may obtain from the items chosen in $\boldsymbol{X}$. Formally, $\sum_{j=1}^N \mathbb{P}\left(\mathcal{M}(\boldsymbol{X}, u_i) = j\right) = \sigma_i(\boldsymbol{X})$.

---

**Algorithm 1:** Shapley Mediator

---

**Input:** A strategy profile $\boldsymbol{X} = (X_1, \ldots, X_N)$ and a user $u_i$
**Output:** An element from $\{\emptyset, X_1, \ldots, X_N\}$

**1** Pick $Y$ uniformly at random from $(0, 1)$
**2 if** $Y > \max_{j \in [N]} \sigma_i(X_j)$ **then**
**3** $\quad$ return $\emptyset$
**4 else**
**5** $\quad$ Return an element uniformly at random from $\{X_j \mid j \in [N], \sigma_i(X_j) \geq Y\}$

---

Efficiency (for brevity, **EF**) binds player payoffs with the maximum satisfaction level of $u_i$ from the items chosen by the players under $\boldsymbol{X}$. It is well known [15, 35] that the Shapley value is uniquely characterized by properties equivalent to **F** and **EF**, when stated in terms of cooperative games. It is therefore obvious that the Shapley mediator satisfies **EF**. [4] Thus, one would expect that the Shapley mediator will be the only mediator that satisfies **F** and **EF**. This is, however, not the case: consider a mediator that runs TOP w.p. $\sigma_i(\boldsymbol{X})$ and NONE otherwise. Clearly, it satisfies **F** and **EF**. In fact, given a mediator $\mathcal{M}$ satisfying **F** and **Complete**, we can define $\mathcal{M}'$ such that

$$\mathbb{P}\left(\mathcal{M}'(\boldsymbol{X}, u_i) = j\right) = \mathbb{P}\left(\mathcal{M}(\boldsymbol{X}, u_i) = j\right) \cdot \sigma_i(\boldsymbol{X}), \tag{3}$$

thereby obtaining a mediator satisfying **F** and **EF**. The question of uniqueness then arises: is **S** derived by satisfying **F** and **EF**? Or even more broadly, are there mediators that satisfy **F**, **S** and **EF** besides the Shapley mediator? Had the answer been yes, this recipe for generating new mediators would have allowed us to seek potentially better mediators, e.g., one satisfying **F**, **S** and **EF** while maximizing user utility. However, as we show next, the Shapley mediator is unique in satisfying **F**, **S** and **EF**.

**Theorem 4.** *The only mediator satisfying **F**, **S** and **EF** is the Shapley mediator.*

## 6 Implications of strategic behavior

In this section we examine the implications of strategic behavior of the players on their payoffs and user utility. Comprehensive treatment of the integration of multiple stakeholders into recommendation calculations was discussed only recently [9], and appears to be challenging. As our work is concerned with strategic content providers, it is natural to consider the Price of Anarchy [24, 34], a common inefficiency measure in non-cooperative games.

### 6.1 Player payoffs

The Price of Anarchy, herein denoted $PoA$, measures the inefficiency in terms of social welfare, as a result of selfish behavior of the players. Specifically, it is the ratio between an optimal dictatorial scenario and the social welfare of the worst PNE. Formally, if $E_{\mathcal{M}} \subseteq \prod_j \mathcal{L}_j$ is the set of PNE profiles induced by a mediator $\mathcal{M}$, then $PoA_{\mathcal{M}} = \frac{\max_{\boldsymbol{X} \in \prod_j \mathcal{L}_j} V(\boldsymbol{X})}{\min_{\boldsymbol{X} \in E_{\mathcal{M}}} V(\boldsymbol{X})} \geq 1$. We use the subscript $\mathcal{M}$ to stress that the $PoA_{\mathcal{M}}$ depends on the mediator, through the definition of social welfare function $V$ and player payoffs. Notice that the $PoA$ of a mediator that does not satisfy **S** can be unbounded, as a PNE may not exist. Quantifying the $PoA$ can be technically challenging; thus we restrict our analysis to $PoA_{\text{SM}}$, the $PoA$ of the Shapley mediator.

**Theorem 5.** $PoA_{\text{SM}} \leq \frac{2N-1}{N}$, *and this bound is tight.*

Hence, under the Shapley mediator the social welfare of the players can decrease by at most a factor of 2, when compared to an optimal solution.

### 6.2 User utility

We now examine the implications of using the Shapley mediator on the users. For that, we shall assume that the utility of a user from an item is his satisfaction level from that item. Namely, when

item $l$ is displayed to $u_i$, his utility is $\sigma_i(l)$. As a result, the expected utility of the users under the strategy profile $\boldsymbol{X}$ and a mediator $\mathcal{M}$ is defined by

$$U_{\mathcal{M}}(\boldsymbol{X}) = \sum_{i=1}^{n} \sum_{j=1}^{N} \mathbb{P}\left(\mathcal{M}(\boldsymbol{X}, u_i) = j\right) \sigma_i(X_j) + \sum_{i=1}^{n} \mathbb{P}\left(\mathcal{M}(\boldsymbol{X}, u_i) = \emptyset\right) \sigma_i(\emptyset).$$

Note that the first term results from the displayed items, and the second term from the plain content of the app (displaying no item at all). To quantify the inefficiency of user utility due to selfish behavior of the players under $\mathcal{M}$, we define the *User Price of Anarchy*,

$$UPoA_{\mathcal{M}} = \frac{\max_{\mathcal{M}', \boldsymbol{X} \in \Pi_{j=1}^{N} \mathcal{L}_j} U_{\mathcal{M}'}(\boldsymbol{X})}{\min_{\boldsymbol{X} \in E_{\mathcal{M}}} U_{\mathcal{M}}(\boldsymbol{X})}.$$

The $UPoA$ serves as our benchmark for inefficiency of user utility. The nominator is the best possible case: the user utility under any mediator $\mathcal{M}'$ and any strategy profile $\boldsymbol{X}$. The denominator is the worst user utility under $\mathcal{M}$, where $E_{\mathcal{M}}$ is again the set of PNE profiles induced by $\mathcal{M}$. Note that the nominator is independent of $\mathcal{M}$. We first treat users as having zero satisfaction when only the plain content is displayed, i.e., $\sigma_i(\emptyset) = 0$, and consider the complementary case afterwards. The following is a negative result for the Shapley mediator.

**Proposition 2.** *The User PoA of the Shapley mediator, $UPoA_{\mathrm{SM}}$, is unbounded.*

Proposition 2 questions the applicability of the Shapley mediator. An unavoidable consequence of its use is a potentially destructive effect on user utility. While content-provider fairness is essential, users are the driving force of the RS. Therefore, one may advocate for other mediators that perform better with respect to user utility, albeit not necessarily satisfying **S**. If **S** is discarded and a mediator satisfying **Complete** adopted, would this result in better user utility? Unfortunately, other mediators may lead to a similar decrease in user utility due to strategic behavior of the players, so there appears to be no better solution in this regard.

**Proposition 3.** *The User PoA of* TOP, *$UPoA_{\mathrm{TOP}}$, is unbounded.*

Using similar arguments, one can show that $UPoA_{\mathrm{BTL}}$ is unbounded as well.

In many situations, it is reasonable to assume that when no item is displayed to a user, his utility is 1. Namely, $\sigma_i(\emptyset) = 1$ for every user $u_i$. Indeed, this seems aligned with the ads-in-apps model: the user is interrupted when an advertisement is displayed. We refer to this scenario as the *optimal plain content* case. From here on, we adopt this perspective for upper-bounding the $UPoA$. Observe that user utility is therefore maximized when no item is displayed whatsoever. Nevertheless, displaying no item will also result in zero payoff for the players. Here too, $UPoA_{\mathrm{TOP}}$ is unbounded, while $UPoA_{\mathrm{NONE}} = 1$. The following lemma bounds the User $PoA$ of the Shapley mediator.

**Lemma 1.** *In the optimal plain content case, it holds that $UPoA_{\mathrm{SM}} \leq 4$.*

In fact, numerical calculations show that $UPoA_{\mathrm{SM}}$ is bounded by 1.76, see the appendix for further discussion.

# 7 Discussion

Our results are readily extendable in the following important direction (which is even further elaborated in the appendix). In many online scenarios, content providers typically customize the items they offer to accommodate specific individuals. Indeed, personalization is applied in a variety of fields in order to improve user satisfaction. Specifically, consider the case where each player may promote a set of items, where different items may be targeted towards different users, and the size of this set is determined exogenously (e.g., by her budget). In this case, a player selects a set of items which she then provides to the mediator. Here the Shapley mediator satisfies **F** and **S**; the game induced by the Shapley mediator is still a potential game, and the computation of the Shapley mediator still takes linear time.

# Acknowledgments

This project has received funding from the European Research Council (ERC) under the European Union's Horizon 2020 research and innovation programme (grant agreement n° 740435).

## Footnotes

[1]For a formal definition of the user utility, see Subsection 6.2.

[2]One may require the convergence of any better-response dynamics, thereby allowing the players to learn the environment. In Section 3 we show that our solution satisfies this stronger notion of stability as well.

[3]Perhaps excluding profiles $X$ where $\sigma_i(X) = 0$. We allow $\mathcal{M}$ to behave arbitrarily in this case.

[4] See the proof of Proposition 1 in the appendix. **Leader Monotonicity**, as opposed to the other fairness properties, is not one of Shapley's axioms but rather a byproduct of Shapley's characterization.

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
