[Reviews · NeurIPS 2018]

Reviewer 1



This paper studies a game design problem. There are U users and P players. Each player has a set of possible of actions. Each action of a user gives a certain utility to each player. There is a "mediator" that, upon receiving a profile of actions from all the players, will choose which action to display for each user. Thus the mediator algorithm defines the game, and each player gets a utility of 1 when their action is displayed to a user. The question is can we develop a "fair" mediator algorithm that will result in the underlying game to have a pure Nash equilibrium, where fairness includes intuitive properties like for each user, the highest utility action should be chosen with strictly higher probability than the rest of the actions etc. There is another property called completeness, which states that each user should be displayed an action with probability 1 (and not lesser than 1). The authors show that no game can simultaneously satisfy fairness, completeness and have a pure strategy Nash equilibrium. They propose a game that chooses each action with probabilities proportional to the Shapley value suggested increments and show that this satisfies fairness and has a pure strategy Nash. They then propose to study the price of Anarchy, both for user utility and player utility. The Shapley mediator algorithm has a poa of 2 for the player utility and unbounded for the user utility model. My concerns about this paper are I am not convinced that in any of the proposed applications, existence of an equilibrium is an essential concern. In particular, the actions in all recommendation systems are suggested content, and the suggested content is going to vary quite frequently --- no content is going to be shown more than a small fixed number of times to any user (I think of news articles, ads etc.). That being the case why is equilbrium a main concern? Even if we are concerned with the same content being shown often, we still need to assume that the same competing contents are also present in all the repeated interactions. If one forgets the equilibrium, the TOP algorithm has fairness and completeness, has a player POA of 1 and user POA unbounded. It's a much simpler algorithm to implement. Overall, I am not convinced that theoretical analysis of the equilibrium is the right problem to study in this space. The game itself is quite cute though. Also, for a NIPS-like conference, absence of even a tangential connection to any form of Machine Learning is a minus. ************************************* After author response: Author response quote: "In most (and perhaps even all) forms of multi-agent interaction in the Web, mechanism design is carried out to mitigate agent incentives." ---- Please be specific. I am aware that game theory and equilibrium play a central role in many settings, ad auctions being one of the primary examples. There are many relevant questions for equilibrium's applicability. But one of the most important is: how often is the *same* game repeated over time, for things like repeated best response to make sense? First, in personalized offer case, whenever the set of possible actions L_j available to player j is reasonably small (e.g. the set of possible news articles a publisher can promote on any given day is not huge), then player j submitting to the mediator the whole set L_j as their strategy X_j would mean that there are no incentives in the game: for each user i, the mediator algorithm (TOP) will just pick the player j whose best item's satisfaction is highest among all players for user i. There is no room for any incentives here, and the mediator obtains the maximum user welfare as well. The authors mitigate this uninteresting situation by saying that there is a budget k_j on the maximum number of actions that can be a part of the strategy of any player j. Personalization is an important aspect of RSs (as reviewer 2 points out as well), and specifying the regime in which the game is interesting/relevant is essential. For example, the paper begins with Alice and Bob examples, which might suggest that the game is interesting even at a fully personalized level, but it is not. Second, even as an impersonal game (or a semi-personal game, where we put a limit on the set of usable actions as the authors suggest): the relevant question for equilibrium applicability, has to be discussed. How often is the same game repeated over time? Same game means, the payoffs are more or less the same for any given strategy profile. For a given strategy profile consisting of the action chosen (or the set of actions chosen as in Appendix C) by every player, will the user payoffs (which ultimately determines how any mediator algorithm will pick the player item to show, and hence determine the player playoffs, which is what is the game's payoffs here) remain the same over multiple interactions? Even across a day, Alice might want not want to see too much of financial news, and may perhaps derive more utility out of knowing about local businesses via their ads. Therefore in the repeated RS game, the user payoffs most likely changes over time. So while encountering Alice the second time, the payoff she derives (which influences what the mediator chooses, and hence influences the player payoffs) is very different from the first round interaction. Hence the planned response for this interaction should also be different. One can counter this by saying that each user provides a content opportunity only a few times a day, but the sum total of all users is huge, and hence it is a repeated game --- this is fallacious though: the sum total of all users together determine one "round" of this game because the combined payoffs from showing content to all users is what determines the player payoffs in the game. With such a model of player payoffs, a repeated game in the RS context necessarily refers to same user visiting multiple times. Therefore the payoff (and hence the game) also necessarily varies over these repetitions. Circumventing this situation by decomposing player payoffs across users (i.e., by making personalized offers for each user) will make the game uninteresting for reasons described in the previous paragraph. On the other hand, consider the same question for the GSP example. Here, the advertiser payoff can be determined for each user interaction separately (there is ample room for strategizing even when considering a single user, as opposed to the RS game that has no incentives in the personalized user case). Thus for a given keyword, repeatedly interacting with different users, are all repeated plays of the same game, with the same advertiser payoffs (or nearly same, or proportionally same if some users just click less often on ads). Thus, things like repeated best response etc. make direct sense here. And indeed it was in one such repeated best response that the famous saw-tooth bidding behavior happened in the first price auction that was previously used by search engines, prompting the move to the generalized second price auction. In fact other repeated games like selfish routing also have this property that when all players use the same strategy profile, the payoffs remain the same. It doesn't depend on whether the same routes were already used by all car drivers yesterday! If everyone drives the same route today, the payoffs remain the same. There are other concerns like how accurately can the players determine their payoffs for any given strategy, or even get a sense of how many players one is competing with. All these things are hard/vague in the RS game. ****************************************** Author response quote: "Using a TOP-like approach for search/recommendation engines (the Probability Ranking Principle, a core concept in information retrieval) has lead to the creation of a 60 Billion USD market (in 2015, in the US alone) named Search Engine Optimization (SEO)." ---- a multi-billion dollar market for SEO might have been created regardless, partly because there are numerous so-called "white hat SEOs" that could legitimately improve the visibility and ranking of a website. There is a multi-billion dollar market just for optimizing bidding strategies in display ad auctions, which have predominantly used second price auctions for a long time (although that is changing now). What is the need for optimizing bidding strategies in a second price auction which is truthful? Perhaps the bidder is not able to predict the click-through-rate themselves, and uses the help of a demand-side-platform. Perhaps, the bidder is not able to optimally manage their budget, etc. Likewise, there are numerous reasons for an SEO market to exist, and saying "TOP-like approach for search/recommendation engines has led to the creation of a 60 Billion USD market" is quite inaccurate. ****************************************** All that said, I still find the paper game theoretically beautiful. The way the authors design the game deserves to be published. But given the above, and the fact that the connection to learning is quite tenuous, at least NIPS doesn't seem ideal to me. I wouldn't object publication though.

Reviewer 2



The paper presents an interesting formulation for recommender systems, with a special focus on the utility of the content providers. They propose the Shapley mediator to allocate items (chosen by content providers or players) to users in order to ensure that the player's rewards are fairly and stably allocated. They also propose a linear time algorithm to implement the mediator which would make it very feasible for usage in real life scenarios. The strengths of the paper: - Exceptionally well written. Very clear exposition of the key ideas. - Novel ideas, and the linear time algorithm is simple, elegant and necessary for broader usage. - Mathematically and logically sound. A couple of things that could be better motivated. - (Major) Why is it necessary to have an option not to show an item? Can the plain content not be treated as another player (if yes, what would be its profile, and if not, why not?) - (Minor) In sections 3 and 4, there is initially some lack of clarity of how the Shapley value and the probability of the Shapley mediator are related. Only point where the writing could have been clearer. Areas for improvement - One critical point which only gets mentioned at the very end (in the discussion section) is that the profile of a player is generally not the same for all users. It is instead personalized for almost all recommender systems. While the authors allude to having solved that problem too, the paper does not contain any of that information. - The user utility PoA being unbounded while being theoretically true is practically quite unsatisfactory. For such a mediator to be useful in practice, it may need to balance player payoffs with user utilities. One general comment: - The format of the paper and the volume of accompanying material (which are somewhat critical but omitted from the paper) make it slightly more suitable for a journal publication. Finally, my confidence score is medium because I am not familiar with related work in the area, although I am fairly confident that I understood most of what was presented in the (main) paper.

Reviewer 3



Takes a game implementation approach to recommendation systems design. Model: there are n advertisers. A user shows up, and then each picks an ad to push to a "mediator" (e.g. content provider). The user's satisfaction with various ads is assumed known. Player utilities correspond to the probability that their ad is shown. Note that choosing a mediator in effect defines the payoff matrices of the players in a suitable game (players = advertisers, strategies = which ad to push). Thus we have a game design question. The submission shows some issues with simple mediators (e.g., violating fairness in some sense) and proposes a novel mediator based on the Shapley value. (Where given player strategies and a user, the relevant cooperative game is defined by v(S) where v(S) denotes the given user's maximum satisfaction by any of the adds pushed by players in S). Because the Shapley value has so many nice properties, so does this induced game (e.g., existence of pure-strategy Nash equilibria). Also, the special form of the game allows Shapley values to be computed efficiently (usually computing them is #P-hard). I found this an innovative application of game-theoretic techniques.